

# Vet the journal before you submit: turnaround times of journals publishing in zoological medicine and related fields

Brendan Runde[1,2] and Craig Harms[3]

[1] Department of Applied Ecology, College of Agriculture and Life Sciences, Center for Marine Sciences and Technology, North Carolina State University, Morehead City, NC, United States of America
[2] The Nature Conservancy, Charlottesville, VA, United States of America
[3] Department of Clinical Sciences, College of Veterinary Medicine, Center for Marine Sciences and Technology, North Carolina State University, Morehead City, NC, United States of America

## ABSTRACT

Many factors influence selection of a target journal for publishing scientific papers, including "fit" within the journal's scope, acceptance rate, readership, open access options, submission and publication costs, journal quality, and timeliness of publication. Timeliness of publication can be a critical factor affecting career development, but many journals are not transparent about turnaround times. Here we evaluated 49 journals publishing papers in zoological medicine and related fields between 2017 and 2022, and aggregated and examined distributions of turnaround time of journals that publicly provided the requisite data, in order to aid authors in selecting target journals that best meet their needs. Of 49 journals evaluated, 39 provided necessary dates for reconstructing turnaround times. Of these, median times to acceptance ranged from 37 to 338 days, and median times to publication ranged from 41 to 403.5 days. The percentage of papers published in greater than 1 year ("slow") ranged from 0 to 57.1%, while the percentage of papers published in under 6 months ("timely") ranged from 0.8 to 99.8%. Acceptance rates and times to first decision were available for only 22% and 20%, respectively, of journals evaluated. Results may prove useful for authors deciding where to submit their works, depending on how they prioritize the many factors involved.

## INTRODUCTION

Selecting a target journal for a scientific manuscript can be a difficult decision. Important factors include the paper's "fit" within the journal's scope, likelihood of acceptance, target readership, open access options, submission and publication costs, a measure of journal quality such as impact factor, less easily quantified qualities such as review quality and editorial management, and timeliness of publication. Unfortunately, many journals and publishers are less-than-transparent about some of these factors—such as turnaround time and acceptance rate—leaving authors to rely on anecdotal and limited experiential information.

Corresponding author
Craig Harms, caharms@ncsu.edu

Choosing a journal that does not meet one's needs and expectations can be professionally detrimental, as producing peer-reviewed research within set time frames is important to career advancement in many fields. For instance, in zoological medicine (construed broadly to include zoo, wildlife, aquatic, and exotic animal medicine) and other disciplines, publishing is important for students and early career professionals to compete successfully for internships and residencies, credentialing for specialty boards, job applications, and promotion and tenure decisions. Training programs often set expectations for number of publications within the time frame of the internship or residency as a condition of awarding certificates of completion, and academic institutions have expectations for scholarly productivity within a defined timeline in order to be considered for promotion and tenure, which can be particularly challenging for faculty with clinical service responsibilities to meet (*San Miguel, 2019*). Because early career research results often naturally manifest towards latter stages of student years, internship and residency programs, and pre-tenure time lines, prolonged turnaround time for a paper may cause delays in professional benchmarks, such as specialty board credentialing and tenure. Time to acceptance is a critical factor for individuals compiling credentials packets for the American College of Zoological Medicine (ACZM) examination, as credentialling requires that the applicant "be first author on at least three (3) publications relevant to the field of zoological medicine in refereed journals," and "the manuscript must be fully accepted for publication prior to the deadline for applying for the certification examination" (*American College of Zoological Medicine, 2022*). Similar conditions apply for individuals pursuing board certification in the European College of Zoological Medicine—Zoo Health Management (ECZM (ZHM)), for which the applicant must be author of "three (3) original peer-reviewed scientific papers in a well-established internationally refereed scientific journal…," and "must be the principal author" of at least two of those (*European College of Zoological Medicine, 2020*). These conditions may place a premium on expeditious manuscript review at any career stage for an individual pursuing ACZM or ECZM (ZHM) board certification, but particularly so for third-year residents in zoological medicine programs aiming to sit one of these board examinations immediately following their residency.

Inaccessible data on manuscript turnaround times hinder informed journal submission decisions by authors in time-sensitive situations. Some journals that publish papers in zoological medicine and related fields provide per-paper publication histories (*i.e.,* date received, date accepted, and date published) that allow for the determination of individual turnaround times. Aggregating turnaround times for many papers would allow for the generation of per-journal statistics that could be used by authors to refine their decision of which journal to target. Turnaround time statistics for over 80 journals that publish papers in fisheries science were recently provided (*Runde, 2021*). Here, we evaluate 49 journals that publish papers in zoological medicine and related fields, including those on the ACZM Annotated Suggested Reading List (*American College of Zoological Medicine, 2022*) and the ECZM (ZHM) reading list (*European College of Zoological Medicine, 2020*), with the same goal as the fisheries science paper (*Runde, 2021*): to aid authors in selecting target journals that best meet their needs and expectations.

## MATERIALS & METHODS

We developed a list of journals for inclusion in this study based on all journals on the ACZM and ECZM (ZHM) reading lists from 2021 and 2022; these lists were supplemented with related journals in the field based on the authors' knowledge (in particular, CAH, diplomate of ACZM and ECZM (ZHM), Past President of ACZM, 30 years in the field). The final list was comprised of 49 journals, including some that do not publish exclusively in the field of zoological medicine (*e.g., Journal of the American Veterinary Medical Association*) or even of veterinary medicine (*e.g., PeerJ*; Table 1), but which nevertheless publish relevant papers and in several instances are included on ACZM and ECZM (ZHM) reading lists. Some higher impact journals that occasionally contain papers of zoological medicine and related field interest (*e.g., Science, Nature*) are not included here because of their broader scope and their inclusion in a prior paper (*Runde, 2021*), while some journals included in that study more relevant to the current paper are included with updated data.

For each journal, we obtained the 2020 impact factor, the most recent year for which it was available for all journals evaluated (*Resurchify, 2020*). Impact factor is calculated as the number of citations received in a given year by all papers published in that journal during the previous 2 years, divided by the number of papers published in that journal in that timespan. Due to its susceptibility to manipulation, impact factor is considered an imperfect metric of journal quality (*Ioannidis & Thombs, 2019*; *Seglen, 1997*), but is still widely relied on by many authors (*Archambault & Larivière, 2009*; *Smith, 2006*).

For each journal, we accessed the webpage and/or PDFs of recently published papers and located publication history information (*i.e.*, the dates the paper was received, accepted, and published) if it was available. Dates were tabulated for each paper, generally back to the beginning of 2018. Where possible, we excluded publications that were not original research (*e.g.*, reviews, brief communications, editorials, errata), on the assumption that such documents have inherently different turnaround times.

We examined distributions of time-to-acceptance (*date accepted–date received)* and time-to-publication (*date published–date received*) for each journal where information was available. Some papers list multiple publication dates (*i.e., date published online* and *date published in issue*)—we always used whichever publication date came first (generally, *date published online*).

Some papers reported inconceivably short time-to-acceptance or time-to-publication (*e.g.*, received to accepted in 0 d). It is extremely unlikely that a peer-reviewed paper could legitimately be submitted and accepted on the same day. In fact, the authors consider that any acceptance or publication occurring in under 14 d is likely not reflective of the paper's true timeline. To that end, we eliminated from further analysis any papers accepted or published in 14 d or less from the date received (although we report the proportion excluded on that basis for each journal).

We generated summary data in R 4.1.1 (*R Core Team, 2021*) for each journal with publication history. Specifically, we focused on median time-to-acceptance and median time-to-publication for each journal. We also evaluated each journal for the proportion of papers published in under 6 months (considered "timely") and the proportion of papers

Runde and Harms (2023), *PeerJ*, DOI 10.7717/peerj.15656

**Table 1  Summary data for full suite of 49 journals examined.**

| Journal | IF | 2021 ACZM | 2022 ACZM | 2022 ECZM ZHM | Turnaround time obtained? | Acceptance rate | Days to first decision |
|---|---|---|---|---|---|---|---|
| American Journal of Veterinary Research | 0.9 | Core | Core | X | Yes | | |
| Animals | 3.1 | | | | Yes | | |
| BMC Veterinary Research | 2.7 | | | | Yes | | |
| Canadian Veterinary Journal | 0.6 | | | | Not Reported | | |
| Chelonian Conservation and Biology | 1.1 | | | | Yes | | |
| Conservation Biology | 6.7 | Additional | | | Yes | 0.15 | 55 |
| Conservation Physiology | 2.8 | | | | Yes | | |
| Diseases of Aquatic Organisms | 1.7 | Additional | Additional | | Yes | | |
| Emerging Infectious Diseases | 9.9 | Additional | Additional | | Not Reported | | |
| Fish and Shellfish Immunology | 4.3 | | | | Yes | 0.44 | 35 |
| Frontiers in Veterinary Science | 3.2 | | | | Yes | | |
| Herpetological Conservation and Biology | 1.1 | | | | Yes | | |
| Journal of Aquatic Animal Health | 2.5 | Additional | Additional | | Yes | | |
| Journal of Avian Medicine and Surgery | 0.4 | Core | Core | X | Not Reported | | |
| Journal of Exotic Pet Medicine | 0.5 | Additional | Additional | | Not Reported | 0.41 | 21 |
| Journal of Fish & Wildlife Management | 0.8 | | | | Yes | | |
| Journal of Fish Diseases | 2.6 | | | | Yes | | |
| Journal of Herpetological Medicine and Surgery | NL | Core | Core | X | Not Reported | | |
| Journal of Invertebrate Pathology | 2.8 | | | | Yes | 0.37 | 42 |
| Journal of Medical Primatology | 0.7 | | | | Yes | | |
| Journal of Small Animal Practice | 1.4 | Additional | Additional | | Yes | | |
| Journal of the American Association for Laboratory Animal Science | 1.4 | Additional | Additional | | Yes | 0.6 | |
| Journal of the American Veterinary Medical Association | 0.7 | Core | Core | X | Not Reported | | |
| Journal of Veterinary Diagnostic Investigation | 1.4 | | | | Not Reported | | |
| Journal of Veterinary Medical Education | 1.0 | | | | Not Reported | | |
| Journal of Veterinary Pharmacology and Therapeutics | 1.5 | | | | Yes | | |
| Journal of Wildlife Diseases | 1.6 | Core | Core | X | Yes | | |
| Journal of Wildlife Management | 2.3 | Additional | Additional | | Yes | | |
| Journal of Zoo and Aquarium Research | NL | | Additional | X | Yes | | |
| Journal of Zoo and Wildlife Medicine | 0.8 | Core | Core | X | +/-, no "dates received" | | 68 |
| Marine Mammal Science | 2.1 | | | | Yes | | |

Runde and Harms (2023), *PeerJ*, DOI 10.7717/peerj.15656

**Table 1** (*continued*)

| Journal | IF | 2021 ACZM | 2022 ACZM | 2022 ECZM ZHM | Turnaround time obtained? | Acceptance rate | Days to first decision |
|---|---|---|---|---|---|---|---|
| PeerJ | 3.0 | | | | Yes | 0.42 | 30 |
| PlosOne | 3.6 | | Additional | | Yes | 0.47 | 43 |
| Research in Veterinary Science | 2.5 | | | | Yes | | |
| The Journal of Veterinary Medical Science | 1.1 | | | | Yes | | |
| The Veterinary Journal | 2.6 | | | | Yes | | |
| Theriogenology | 2.9 | | | | Yes | 0.33 | 38 |
| Veterinary Anaesthesia and Analgesia | 1.4 | | | | Yes | | |
| Veterinary Clinical Pathology | 0.8 | | | | Yes | | |
| Veterinary Medicine International | 1.7 | | | | Yes | 0.19 | |
| Veterinary Microbiology | 3.2 | | | | Yes | 0.21 | 34 |
| Veterinary Ophthalmology | 1.5 | | | | Yes | | |
| Veterinary Parasitology | 2.9 | | | | Yes | | |
| Veterinary Pathology | 2.5 | | | | Not Reported | | |
| Veterinary Quarterly | 7.4 | | | | Yes | 0.25 | 24 |
| Veterinary Radiology and Ultrasound | 1.1 | | | | Yes | | |
| Veterinary Record | 0.4 | Additional | Additional | | Yes | | |
| Veterinary Record Open | 1.6 | | | | Yes | | |
| Zoo Biology | 1.6 | Additional | Additional | X | Yes | | |

**Notes.**

Full suite of 49 journals examined, including impact factor (IF), designation as Core or Additional in the suggested reading list for the American College of Zoological Medicine (ACZM) for both 2021 and 2022, inclusion in the suggested reading list of the European College of Zoological Medicine –Zoo Health Management (ECZM ZHM) for 2022, whether or not turnaround time was obtained, acceptance rate (if available), and days to first decision (if available). NL, not listed.

published in over 12 months (considered "slow"), though we suggest that these metrics be used as general guidance as these cutoffs are arbitrary.

## RESULTS

Of 49 journals included, 39 reported *date received, date accepted,* and *date published* on at least a portion of their published papers (Table 1). One journal (*Journal of Zoo and Wildlife Medicine*) reported *date accepted* and *date published* but not *date received.* The remaining nine journals generally reported only *date published.* One of these (*Journal of the American Veterinary Medical Association*) claims an average time from submission to publication of less than 100 d (*Fourtier, 2022*), but does not report *date received* or *date accepted.* From these 39 journals, we collected information on 29,054 individual papers.

For the 39 journals that provided requisite publication histories, median times to acceptance ranged from 37 to 338 d and median times to publication ranged from 41 to 403.5 d (Table 2, Fig. 1). Most of these journals published zero papers in under 14 d from receipt, although 13 journals published between 0.1% and 3.1% of papers in this extremely rapid timeframe (these papers were not included in analyses; see Methods). The percentage of papers for which publication took over 1 year ranged from 0 to 57.1%, while the percentage of papers published in under 6 months ranged from 0.8 to 99.8%. Most journals published at least a few papers that had extremely lengthy turnaround times, ranging up to 1,700 d (Fig. 1).

Acceptance rates and times to first decision were obtained from journal websites or a prior publication[3] for 11 and 10 journals, respectively. Reported acceptance rates ranged from 15% to 60% and reported times to first decision ranged from 21 d to 68 d (Table 1). We did not conduct further analyses with these data given their scarcity among journals examined.

## DISCUSSION

Over 20% of journals examined do not report one or more dates necessary for reconstructing turnaround times. In the interest of transparency in the scientific publication process, we encourage the publishers and editorial staff of these journals (detailed in Table 1) to report this information. Some progress in this direction may already be occurring since the time frame over which data were collected for this study. Similarly, we encourage more journals to be transparent about acceptance rates and times to first decision. Only 22% of journals considered here report acceptance rates and only 20% report times to first decision.

Journals in the ACZM and ECZM (ZHM) reading lists are selected by diplomates of the respective Colleges' examination committees for their relevance to zoological medicine as determined through formal job task analyses and examination validation exercises. For journals on the reading lists that are not exclusively focused on zoological medicine (*e.g., American Journal of Veterinary Research, Emerging Infectious Diseases, etc.*), only content relevant to zoological medicine is considered as potential examination material. Here we assume that turnaround times do not substantially differ among papers of different

**Table 2** Publication histories for the 39 journals that provided requisite information.

| Journal | % Under 14 d | Number Of articles | Start date | End date | Median days to acceptance | Median days to publish | % Over 1 year | % Under 6 months |
|---|---|---|---|---|---|---|---|---|
| American Journal of Veterinary Research | 0.4 | 277 | 1/1/2018 | 9/1/2021 | 108 | 337 | 36.1 | 1.4 |
| Animals | 0.5 | 4,605 | 9/28/2020 | 6/20/2022 | 37 | 41 | 0.0 | 99.8 |
| BMC Veterinary Research | 0 | 1,369 | 1/2/2018 | 6/27/2022 | 184 | 202 | 11.0 | 41.6 |
| Chelonian Conservation and Biology | 0 | 209 | 1/1/2015 | 12/3/2021 | 140 | 337 | 44.5 | 6.7 |
| Conservation Biology | 0 | 361 | 3/31/2017 | 5/16/2021 | 199 | 234 | 14.4 | 32.1 |
| Conservation Physiology | 0.2 | 618 | 1/5/2016 | 7/7/2022 | 144 | 182.5 | 7.4 | 47.6 |
| Diseases of Aquatic Organisms | 2.8 | 436 | 3/22/2018 | 6/23/2022 | 146 | 218 | 7.6 | 30.3 |
| Fish and Shellfish Immunology | 0 | 2,128 | 7/27/2017 | 7/3/2021 | 91 | 95 | 0.4 | 92.6 |
| Frontiers in Veterinary Science | 0 | 2,007 | 4/9/2020 | 6/10/2022 | 71 | 110 | 0.8 | 86.2 |
| Herpetological Conservation and Biology | 0 | 260 | 4/30/2017 | 12/16/2021 | 236.5 | 324 | 41.9 | 9.2 |
| Journal of Aquatic Animal Health | 0 | 74 | 12/18/2017 | 10/11/2021 | 185.5 | 253.5 | 18.9 | 25.7 |
| Journal of Fish and Wildlife Management | 0 | 197 | 2/1/2016 | 4/14/2022 | 211 | 212 | 13.7 | 28.4 |
| Journal of Fish Diseases | 0 | 480 | 7/5/2017 | 2/12/2021 | 62 | 105 | 0.0 | 93.8 |
| Journal of Invertebrate Pathology | 0.2 | 467 | 10/24/2017 | 6/27/2022 | 128 | 132 | 3.2 | 67.9 |
| Journal of Medical Primatology | 0 | 101 | 7/24/2018 | 6/5/2022 | 113 | 149 | 5.0 | 63.4 |
| Journal of Small Animal Practice | 0 | 415 | 7/18/2017 | 8/31/2021 | 186 | 254 | 20.7 | 24.8 |
| Journal of the American Association for Laboratory Animal Science | 0 | 278 | 3/1/2018 | 5/1/2022 | 101 | 280.5 | 10.8 | 5.4 |
| Journal of Veterinary Pharmacology and Therapeutics | 0 | 305 | 4/27/2017 | 5/19/2022 | 120 | 154 | 1.0 | 63.6 |
| Journal of Wildlife Diseases | 0 | 374 | 1/1/2018 | 3/11/2022 | 133 | 307 | 28.9 | 11.0 |
| Journal of Wildlife Management | 0.9 | 530 | 11/5/2016 | 9/21/2021 | 206 | 261 | 22.6 | 18.3 |
| Journal of Zoo and Aquarium Research | 0 | 68 | 1/31/2021 | 7/31/2022 | 338 | 403.5 | 55.9 | 11.8 |
| Marine Mammal Science | 0 | 274 | 8/21/2017 | 3/18/2022 | 289.5 | 340.5 | 41.2 | 7.7 |
| PeerJ | 0 | 2,681 | 2/12/2013 | 7/18/2022 | 124 | 158 | 5.3 | 61.8 |
| PlosOne | 0 | 2,883 | 5/16/2018 | 5/11/2022 | 164 | 194 | 9.2 | 43.7 |
| Research in Veterinary Science | 0.1 | 915 | 5/23/2017 | 7/7/2022 | 160 | 167 | 6.9 | 54.9 |
| The Journal of Veterinary Medical Science | 0 | 1,297 | 11/6/2017 | 3/14/2022 | 103 | 120 | 3.5 | 75.1 |
| The Veterinary Journal | 0 | 7 | 3/2/2022 | 5/11/2022 | 290 | 299 | 28.6 | 28.6 |
| Theriogenology | 3.1 | 1,938 | 6/28/2017 | 7/5/2022 | 136 | 141 | 1.4 | 70.7 |
| Veterinary Anaesthesia and Analgesia | 0 | 368 | 4/17/2017 | 4/29/2022 | 182 | 226.5 | 16.0 | 32.9 |

Runde and Harms (2023), *PeerJ*, DOI 10.7717/peerj.15656

**Table 2** (*continued*)

| Journal | % Under 14 d | Number Of articles | Start date | End date | Median days to acceptance | Median days to publish | % Over 1 year | % Under 6 months |
|---|---|---|---|---|---|---|---|---|
| Veterinary Clinical Pathology | 0 | 119 | 8/17/2018 | 3/30/2022 | 151 | 382 | 57.1 | 0.8 |
| Veterinary Medicine International | 0 | 96 | 11/27/2020 | 7/1/2022 | 117.5 | 137 | 5.2 | 63.5 |
| Veterinary Microbiology | 0.3 | 1,134 | 11/6/2017 | 8/5/2022 | 93 | 98 | 1.5 | 87.4 |
| Veterinary Ophthalmology | 0 | 75 | 2/4/2019 | 5/5/2022 | 166 | 196 | 9.3 | 42.7 |
| Veterinary Parasitology | 0.3 | 714 | 11/2/2017 | 8/4/2022 | 109.5 | 116 | 1.5 | 79.8 |
| Veterinary Quarterly | 1.6 | 61 | 12/1/2014 | 8/8/2022 | 176 | 206 | 9.8 | 39.3 |
| Veterinary Radiology and Ultrasound | 0.3 | 384 | 8/30/2017 | 6/9/2022 | 147 | 224 | 9.6 | 23.2 |
| Veterinary Record | 0 | 198 | 1/6/2018 | 6/14/2022 | 181.5 | 251..5 | 20.2 | 10.6 |
| Veterinary Record Open | 0 | 155 | 1/5/2016 | 8/2/2022 | 150 | 195 | 16.1 | 40.6 |
| Zoo Biology | 0.5 | 196 | 12/6/2017 | 3/7/2022 | 233 | 259 | 20.9 | 29.1 |

**Notes.**

For the 39 journals that provided requisite publication histories: percentage of papers accepted in under 14 days, number of articles analyzed, start and end dates of analyses, median number of days to acceptance and to publication, percentage of papers published in over 1 year from submission (considered prolonged), and percentage of papers published in under 6 months (considered timely).

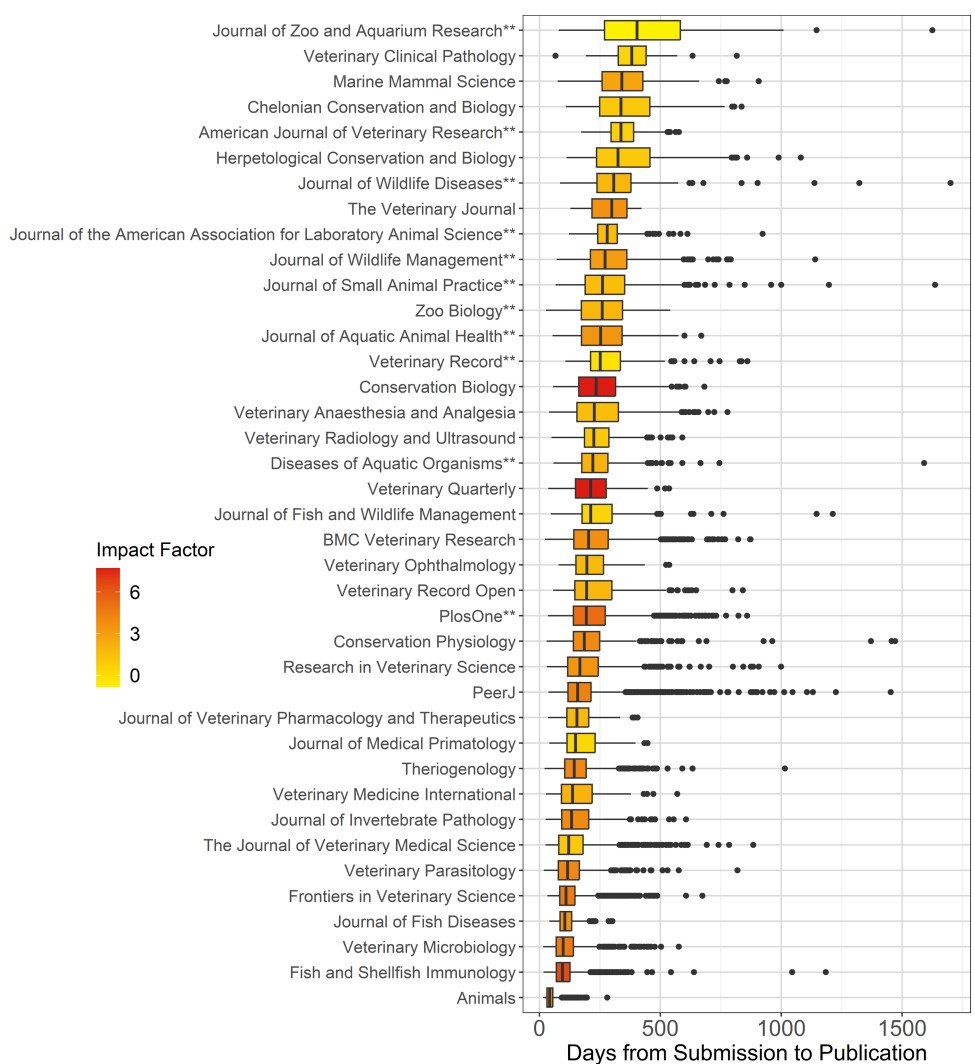

**Figure 1** Box plots showing days from submission to publication for 39 journals that publish papers in zoological medicine and related topics organized in descending order of medians. Central vertical lines represent medians, hinges represent the 25th and 75th percentiles, and lower and upper whiskers extend to either the lowest and highest values respectively, or 1.5 * the inter-quartile range. Black dots represent papers that were outside the 1.5 * inter-quartile range. Boxes are shaded to correspond with 2020 Impact Factor, where darker green represents higher impact. Two asterisks (**) = journals included on either 2022 suggested reading lists of the ACZM or ECZM. Journals on ACZM or ECZM reading lists not included in this figure do not provide requisite data on *date received, date accepted,* and *date published*.

topics in these journals. The designation of journals as primary source material for certification examinations could be expected to put a premium on publishing therein, both by diplomates providing content to build the body of zoological medicine literature for dissemination of knowledge to candidates and for question generation, and by candidates who might then have the happy chance of seeing an examination question sourced from their research. Such journals could also reasonably be expected to have a team of the most knowledgeable and dedicated associate editors and reviewers in the field, who can provide

the most informed peer review to improve the final manuscript. Similar considerations would apply to other veterinary specialty colleges with publication requirements for credentialing, albeit a lesser number of publications required than ACZM.

Other factors frequently take priority when selecting target journals, however. Interests in some of these factors may converge between residents and faculty mentors, and some may diverge. For instance, publication in a higher impact journal may be prioritized by a junior faculty member with a comfortable timeline to promotion and tenure, while her 3rd year resident under pressure to credential for the board examination that year would prioritize time to acceptance and less restrictive acceptance rates. Median impact factor of the core journals of the combined ACZM and ECZM (ZHM) reading lists is only 0.7, with a range from 0 (or no impact factor designated) to 1.5. Neither ACZM nor ECZM (ZHM) take journal impact factor into account in candidate credentialing, and require only that candidates' manuscripts "be published in a refereed journal" (*American College of Zoological Medicine, 2022*) "or" in a well-established internationally refereed scientific journal" (*European College of Zoological Medicine, 2020*). Both organizations elaborate further on what those criteria entail, with inclusion on the suggested reading lists being a favorable factor. For academic promotion and tenure committee deliberations, however, journal impact factor often does play an explicit role (*McKiernan et al., 2019*). Note again, however, that impact factor is an imperfect indicator of journal quality (*Ioannidis & Thombs, 2019*; *Runde, 2021*; *Seglen, 1997*). Longer publication intervals common in zoological medicine journals (*e.g.,* quarterly *versus* monthly, weekly or on a continuous rolling basis) can adversely affect both turnaround times and impact factors. Open access can be a common desire, but high fees for open access can be a limiting factor by cutting into grant or personal funds, and have been implicated in leading to inequitable representation of authors from low-income countries (*Smith et al., 2022*; *Solomon & Björk, 2012*).

Time to first decision may be the most relevant metric of journal editorial and reviewer efficiency, because it is less affected by author responsiveness to reviews. This metric, however, is even less frequently reported than times to acceptance and publication (just 20% in the journals evaluated here). Further, it does not account for quality of reviews. Review quality and unusually rapid turnaround times are concerns that have been raised particularly with respect to mega-journals and special issues with guest editors, where there may be apparent editorial bias and nepotistic behavior (*Scanff et al., 2021*) and varying levels of editorial competence of guest editors compared with professional editors of traditional journals (*Brainard, 2023*; *Ioannidis, Pezzullo & Boccia, 2023*). The Web of Science recently delisted more than 50 journals from its Master Journal List, based on evaluation of 24 measures of quality that include effective peer review (*Brainard, 2023*). Some of the delisted journals come from the major open-access publishers Hindawi and MDPI (*Brainard, 2023*), although not any journals evaluated in the present work as yet. Delisted journals are deprived of an impact factor, which can affect both the journal as a less attractive target for paper submissions, and authors who may be relying on the impact factor metric to bolster their promotion and tenure dossiers.

It is important to acknowledge author responsibilities in ensuring a timely turnaround to publication, including following journal formatting guidelines in the initial submission,

and promptly responding to reviews. Here we assume that timeliness and tardiness by authors was roughly equivalent among journals examined.

In recent years, some journals have rendered "reject and resubmit" decisions unto reviewed papers that might historically have been tasked with "major revisions." Early proponents of this alternative noted that potential benefits could include a quicker turnaround time and slightly better chance of publication (*Range & Tingstrom, 1992*). However, a byproduct of this process is that the resubmitted version would be entered as a new submission with a new starting date. While this practice is not necessarily carried out with intent to artificially depress turnaround time statistics, it introduces bias in studies such as this one. Unfortunately, we cannot account for this bias but encourage journals to use this decision sparingly and never for manipulation.

As noted previously for fisheries journals (*Runde, 2021*), turnaround times, acceptance rates, and impact factors reported in this study are not fixed and can change over time. Further, readers and editors acting upon these results may cause some change. Aggregated data in the current study should therefore be considered baseline information for the timespan evaluated. Results may prove useful to authors deciding where to submit their works, depending on how they prioritize the factors involved. Despite our focus on turnaround times, in deciding where we should submit the current manuscript we also prioritized open access, target readership and journal "fit" of a publication that commonly carries zoological medicine topics. This is in accord with recommendations that authors consider "fit" as the most important factor in deciding where to submit their manuscripts (*Knight & Steinbach, 2008*), followed by other factors that best align with their priorities (*Runde, 2021*; *Solomon & Björk, 2012*).

## CONCLUSIONS

We evaluated 49 journals that publish papers in zoological medicine and related fields, including those on the ACZM Annotated Suggested Reading List and the ECZM (ZHM) Reading List, and analyzed turnaround times for the 39 of those journals publicly providing the requisite data. Additionally, we have aggregated impact factors, and, where publicly available, acceptance rates and times to first decision. Results will aid authors in selecting target journals that best meet their needs and expectations.

## ACKNOWLEDGEMENTS

We thank Nancy Lung for discussions on an earlier version of the manuscript, and reviewers for their comments.

### Funding

No designated funding was provided for this work; general support came from the State of North Carolina (Brendan Runde, Craig Harms). Partial support for publication fees came from the Department of Clinical Sciences Faculty Research Dissemination Fund (Craig

Harms). The funders had no role in study design, data collection and analysis, decision to publish, or preparation of the manuscript.

## Grant Disclosures

The following grant information was disclosed by the authors:
State of North Carolina Department of Clinical Sciences Faculty Research Dissemination Fund.

## Competing Interests

The authors declare there are no competing interests.

## Author Contributions

- Brendan Runde conceived and designed the experiments, performed the experiments, analyzed the data, prepared figures and/or tables, authored or reviewed drafts of the article, and approved the final draft.
- Craig Harms conceived and designed the experiments, analyzed the data, authored or reviewed drafts of the article, and approved the final draft.

## Data Availability

The raw data are available in the Supplementary File.

## Supplemental Information

Supplemental information for this article can be found online at http://dx.doi.org/10.7717/peerj.15656#supplemental-information.

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
