# Peer review of "Vet the journal before you submit: turnaround times of journals publishing in zoological medicine and related fields"

_PeerJ, doi:10.7717/peerj.15656_

## Round 0.1 · original submission · Major Revisions

Both of the reviewers have suggested some changes; hence, my recommendation is to revise.

Reviewer 1 ·

Basic reporting

The inclusion criteria of the journals studied are not clear, and there are some that should not be included. Most of the journals included are from the USA; the spectrum should be broadened to a global level. Authors can select the journal based on other factors, such as the reputation of the journal in their field and the editorial management, review quality, etc. Therefore, I believe that the conclusions cannot say that this paper can help them decide where they should publish.

Experimental design

The inclusion criteria of the journals studied are not clear, most of the journals included are from the USA; the spectrum should be broadened to a global level.

Validity of the findings

Everything is biased and the conclusions made are not shared.

Additional comments

Add Journals from all parts of the world. especially Journals of all disciplines, indexed in Scopus and WOS.

Reviewer 2 ·

Basic reporting

no comment

Experimental design

Dear authors, there are figures in the abstract (which you should also review in the manuscript) that are not clear. They talk about 40 journals, after they evaluated 49, but only 39 were useful.
Please clarify if it was 40 or 49.

Validity of the findings

no comment

Additional comments

Manuscript titled "Vet the journal before you submit: turnaround times of general veterinary and zoological medicine journals (#81812)" for PeerJ.
The authors carried out a very interesting study, it will be a great tool to be able to select an appropriate journal according to some of its characteristics (review time, review time, etc.) for articles related to veterinary medicine in general.
The document is clear and well detailed. The purpose of the study is correctly defined. The details of the methods are understandable. The results are clearly presented. The discussion is pertinent and complete. The conclusions are expressed appropriately and are connected to the original question investigated. The manuscript contains sufficient and adequate references. The figures, tables and supplementary material are good and help to meaningfully analyze and understand the results. The manuscript is interesting among researchers from the many fields of veterinary sciences. Therefore, I would suggest accepting the manuscript for publication on PeerJ.

---

## Round 0.2 · Major Revisions

I note that in your response letter, you explained why you opted to revise the title of the manuscript rather than broaden the search parameters and I accept your choice to do so.

As per the recommendation of reviewer 3, I ask you to revise the manuscript to correct the minor issues and add some discussion regarding recent developments.

Reviewer 1 ·

Basic reporting

The manuscript is the same, with no changes. Tables are not suitable for a scientific journal.

Experimental design

The manuscript is the same, with no changes. Tables are not suitable for a scientific journal.

Validity of the findings

The manuscript is the same, with no changes. Tables are not suitable for a scientific journal.

Reviewer 2 ·

Basic reporting

no comment

Experimental design

Many points were clarified

Validity of the findings

no comment

Additional comments

Dear Editor
I believe that the requested changes were made properly

Reviewer 3 ·

Basic reporting

This is a relatively well written and executed study, albeit somewhat superficial and for a niche audience. I have made some minor comments.

I suggest you explore a bit the published literature and extend the background and the discussion.

line 45 - i think it is important to add that they not only need to publish, but they need a certain number of publications within a given time frame - so there is a large pressure for fast turnover. maybe either here or in the discussion you can bring up predatory journals such as "Animals" which was the fastest on your list, and comment that several MDPI publications have recently lost their impact factor
line 53 insert comma before such
line 61 principal author - first? last? corresponding? all of those count?
line 64 third year residents - third-year residents
lines 65-66 needs a better connection between these two paragraphs
line 95 spell out year
line 117 include version number for R
line 134 The majority of - Most of
line 137 The percentage of papers published in greater than 1 yr ranged - The percentage of articles for which publication took over one year ranged
line 176 is publishing in 0 IF journals penalised by the commitees? needs a bit more discussion
line 180 Open 181 access can be a common desire, but funds for open access can be a limiting factor. - i think this might be a very important factor for decision too - might be interesting to further discuss
lines 190-197 it would be good to include some references for this section. i assume this is a major way journals are "gaming the system"

as mentioned above, it might be worthwhile discussing the extremely short review time of Animals and other MDPI journals and how some of them lost their impact factor recently

it might be better to change the colour of the boxplots from monochromatic to something that is easier to distinguish (with regard to IF)

Experimental design

no comment

Validity of the findings

no comment

Additional comments

no comment

---

## Round 0.3 · accepted · Accept

Both reviewers agreed to accept the manuscript. Hence, my recommendation is to accept.

Reviewer 1 ·

Basic reporting

OK, now all was improved.

Experimental design

Good

Validity of the findings

OK

Additional comments

No more commentaries, the aim is to accept.

Reviewer 3 ·

Basic reporting

I am satisfied with the revised version.

Experimental design

no comment

Validity of the findings

no comment

Additional comments

no comment